# Stress combined with loss of the *Candida albicans* SUMO protease Ulp2 triggers selection of aneuploidy via a two-step process

**Marzia Rizzo[1], Natthapon Soisangwan[2], Samuel Vega-Estevez[1], Robert Jordan Price[3], Chloe Uyl[1], Elise Iracane[1], Matt Shaw[1], Jan Soetaert[4], Anna Selmecki[2], Alessia Buscaino[1]\***

**1** University of Kent, School of Biosciences, Kent Fungal Group, Canterbury Kent, United Kingdom, **2** University of Minnesota, Department of Microbiology and Immunology, Minneapolis, Minnesota, United States of America, **3** Cambridge Crop Research, NIAB, Cambridge, United Kingdom, **4** Blizard Advanced Light Microscopy (BALM), Queen Mary University of London, United Kingdom

\* A.Buscaino@kent.ac.uk

**Data Availability Statement:** Illumina genome sequencing data have been deposited in the Sequence Read Archive (https://www.ncbi.nlm.nih.gov/sra/docs/) under BioProject PRJNA781758.

## Abstract

A delicate balance between genome stability and instability ensures genome integrity while generating genetic diversity, a critical step for evolution. Indeed, while excessive genome instability is harmful, moderated genome instability can drive adaptation to novel environments by maximising genetic variation. *Candida albicans*, a human fungal pathogen that colonises different parts of the human body, adapts rapidly and frequently to different hostile host microenvironments. In this organism, the ability to generate large-scale genomic variation is a key adaptative mechanism triggering dangerous infections even in the presence of antifungal drugs. Understanding how fitter novel karyotypes are selected is key to determining how *C. albicans* and other microbial pathogens establish infections. Here, we identified the SUMO protease Ulp2 as a regulator of *C. albicans* genome integrity through genetic screening. Deletion of *ULP2* leads to increased genome instability, enhanced genome variation and reduced fitness in the absence of additional stress. The combined stress caused by the lack of *ULP2* and antifungal drug treatment leads to the selection of adaptive segmental aneuploidies that partially rescue the fitness defects of *ulp2Δ/Δ* cells. Short and long-read genomic sequencing demonstrates that these novel genotypes are selected via a two-step process leading to the formation of novel chromosomal fragments with breakpoints at microhomology regions and DNA repeats.

## Author summary

In living organisms, the genome is a double-stranded molecule that carries genes. Genes can encode for proteins, the building blocks of life, and protein levels must be tightly balanced. Consequently, genome organisation must be maintained to ensure the balance of genes. However, this is different in microbial organisms such as the human fungal pathogen *Candida albicans*. Indeed *C. albicans* genome structure can change and this fungus

MinION genome sequencing data have been deposited in the Sequence Read Archive under BioProject PRJNA879282. All other relevant data are within the manuscript and its Supporting Information files.

**Funding:** This work was supported by BBSRC (BB/T006315/1 to A.B and S.V.E;), a University of Kent GTA PhD studentships (to M.R.), a University of Minnesota UMR Fellowship with the Bioinformatics and Computational Biology program (to N.S), the National Institutes of Health (R01AI143689) and Burroughs Wellcome Fund Investigator in the Pathogenesis of Infectious Diseases Award (#1020388) to A.S. The funders had no role in study design, data collection and analysis, decision to publish, or preparation of the manuscript.

**Competing interests:** The authors have declared that no competing interests exist.

can live without the right proportion of its genes. This genome plasticity allows the selection of new genome organisations with a combination of genes that drives survival in hostile environments that are found in the human host. Understanding how *C. albicans* and other microbial pathogens thrive in the human host requires understanding how novel genome organisations are selected. We identified the SUMO protease Ulp2 as a regulator of *C. albicans* genome integrity through genetic screening. We discovered that *ULP2* deletion increases *C. albicans'* ability to shuffle its genome and loss of *ULP2* combined with antifungal drug treatment results in the selection of novel genomic organisations. We have sequenced the genome of these novel genotypes and discovered new chromosomal fragments that are selected by two temporally separated steps.

## Introduction

Understanding how organisms survive and thrive in changing environments is a fundamental question in biology. Genetic variation is central to environmental adaptation because it facilitates the selection of fitter genotypes better adapted to a new environment. Different types of genetic changes contribute to genetic variability, including *(i)* whole-chromosome or segmental-chromosome aneuploidy, *(ii)* translocations and *(iii)* mutations [1]. Furthermore, diploid cells can undergo loss of heterozygosity (LOH) driven by mitotic events, such as cross-over, gene conversion or meiotic reversion [1,2]. Whole-chromosome or segmental chromosome aneuploidies have the greatest effect on adaptation as they generate copy number variations (CNVs) of multiple genes. These result in divergent phenotypes which may be selectively advantageous [3].

Genome plasticity–the ability to generate large-scale genomic variation–is emerging as a critical adaptive mechanism in human microbial pathogens that need to adapt quickly to extreme environmental shifts because it provides genetic diversity upon which selection can act [4–8]. One such organism is *Candida albicans*, a common human fungal pathogen and a prevalent cause of death due to systemic fungal infections [9]. *C. albicans* is part of the normal microbiota of most healthy individuals but, in immunocompromised individuals, it is a dangerous pathogen causing a wide range of infections, including life-threatening disseminated diseases [10]. Azole antifungal agents, such as fluconazole (FLC), are the most commonly prescribed drugs for treating *C. albicans* infections [9,11,12].

Several lines of evidence suggest that *C. albicans* genome plasticity provides a competitive advantage under host-relevant stress environments. *C. albicans* is a diploid organism with a heterozygous genome organised into $2 \times 8$ (2n = 16) chromosomes (Chr) [13,14]. Seven chromosomes are designated Chr1 to Chr7 according to size, while one is termed ChrR because it contains the *rDNA* locus [15]. Genomic analysis of clinical isolates reveals that many *C. albicans* strains have large-scale genomic changes including segmental and whole chromosome aneuploidies [16–19]. Furthermore, specific chromosomal variants are selected during host-niche colonisation [17,20–25]. Accordingly, many drug-resistant isolates exhibit karyotypic diversity that can confer resistance due to increased copies of specific genes. For example, CNV for the gene *ERG11* encoding for the target of FLC, lanosterol 14-alpha-demethylase is often observed in drug-resistant isolates [4,16,26–28]. Several studies suggest that *C. albicans* genome instability is not random as it occurs more frequently at specific hotspots which are often repetitive [17,19,23,29,30]. Subtelomeric regions and the *rDNA* locus are among the most unstable genomic sites [17,31]. *C. albicans* subtelomeric regions are enriched in repetitive sequences derived from transposons and protein-coding genes [29,32]. Most notable are the

telomere-associated *TLO* genes, a family of 14 closely related paralogues encoding proteins similar to the Mediator 2 subunit of the Mediator transcriptional regulator [33–35]. Most *TLO* genes are located at subtelomeric regions except *TLO34*, located at an internal locus on the left arm of Chr1 [33]. The *rDNA* locus consists of a tandem array of a ~12 kb unit repeated 50 to 200 times; *rDNA* length polymorphisms frequently occur [14,17].

Despite the clear correlation between genomic variation and environmental adaptation, pinpointing the environmental pressure(s) selecting specific genotypes and understanding how complex karyotypes are formed is often difficult.

In this study, we performed a genetic screening to identify modulators of *C. albicans* genome stability. The screen led to the identification of the *ULP2* gene, encoding for a SUMO protease. *ULP2* deletion causes increased genome instability and enhanced genome variation leading to fitness defects and hypersensitivity to genotoxic agents. We show that loss of *ULP2* combined with exposure to an additional stress (FLC) leads to the selection of multichromosome segmental aneuploidies with adaptive power. Long-read genomic sequencing demonstrates that these novel segmental aneuploidies are selected by a two-step process producing chromosomal fragments with breakpoints at microhomology regions and DNA repeats. Thus, exposure to stress can increase tolerance to unrelated stress by selecting novel complex genotypes.

## Results

### A systematic genetic screen identifies Ulp2 as a regulator of *C. albicans* genotoxic stress response

To identify factors regulating *C. albicans* genome integrity, we utilised a deletion library comprising a subset (674/3000) of *C. albicans* genes that are not conserved in other organisms or have a functional motif potentially related to virulence [36]. As defects in genome integrity lead to hypersensitivity to genotoxic agents [37], the deletion library was screened for hypersensitivity to two DNA damaging agents: Ultraviolet (UV) irradiation which induces formation of pyrimidine dimers [38], and methyl methanesulfonate (MMS), which leads to replication blocks and base mispairing [39]. Genotoxic stress hypersensitivity was semi-quantitatively scored by comparing the growth of treated versus untreated on a scale of 0 to 4, where 0 indicates no sensitivity, and 4 specifies strong hypersensitivity (**Fig 1A**). The screen identified 28 gene deletions linked to DNA damage hypersensitivity (UV or MMS score ≥2). Of those deletion mutants, 9/28 hits show sensitivity to both UV and MMS, 6/28 hits are sensitive only to UV, and 13/28 hits are sensitive only to MMS (**S1 Table**). Functional prediction analysis demonstrated that ~43% of the hits are genes predicted to encode components of the DNA damage response pathway (5/28) or for proteins necessary for cell division (7/28) (**S1 Table**). For example, the top 4 hits of the screen were *GRR1*, *KIP3*, *MEC3* and *RAD18* genes (**S1 Table**). *C. albicans GRR1* and *KIP3* are required for cell cycle progression [40] and mitotic spindle organisation, respectively [41]. Although *C. albicans MEC3* and *RAD18* are uncharacterised, they encode for proteins conserved in other organisms that are universally involved in sensing DNA damage (Mec3) [42] and in DNA post-replication repair (Rad18) [43]. Of the remaining hits, 3/28 genes encode proteins with no apparent ortholog in the two well-studied yeast model systems (*Saccharomyces cerevisiae* and *Schizosaccharomyces pombe*). The last 13 genes encodes for proteins with diverse functions, including stress response (*HOG1*) [44], transcriptional and chromatin regulation (*SPT8*, *SIN3*) [45–47], transport and trafficking (*DUR35*, *NPR2*, *FCY2*, *PEP7*, *VAC14*) [48–52], protein folding (*CNE1*) [53], MAP kinase pathway (*STT4*) [54], phosphatase (*PTC2*) [47], immune evasion (*GPD2*) [55] and cell wall biosynthesis (*KRE5*) [56].

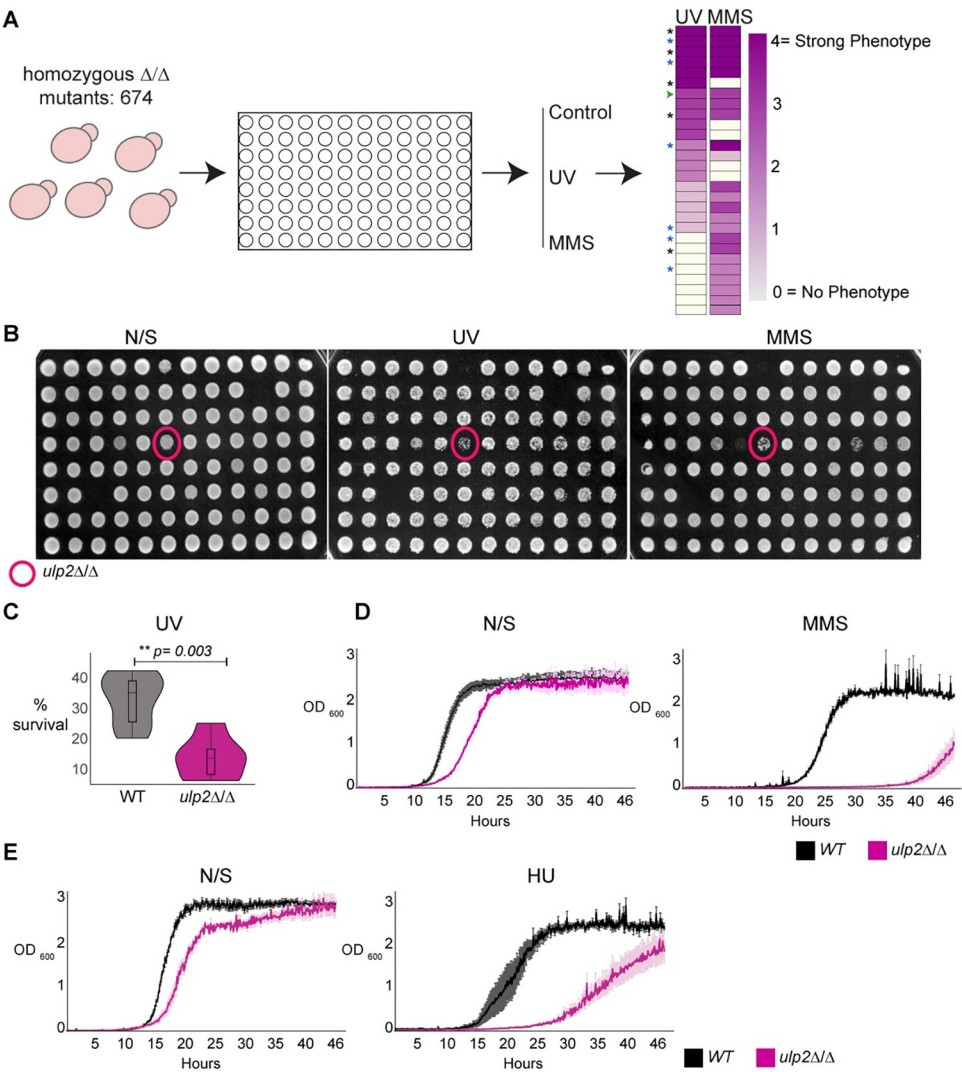

**Fig 1. *ULP2* is a regulator of the *C. albicans* genotoxic stress response. (A)** Schematic representation of the screening strategy. 674 *C. albicans* deletion strains were screened for hypersensitivity to UV and MMS. Hypersensitivity was scored by comparing the growth of treated vs untreated on a scale of 0 (white) to 4 (magenta). Black *: genes encoding for DNA damage and sensing repair pathway components, Blue *: genes encoding for cell division and chromosome segregation machinery, Green arrow: *ulp2Δ/Δ* **(B)** Data for a plate containing *ulp2Δ/Δ* strain (magenta circle). Growth on Non-selective (N/S) media or following UV and MMS treatment is shown **(C)** Colony-forming unit assay (% survival) of UV treated WT and *ulp2Δ/Δ* strain. Statistical analysis was performed using the Kruskal-Wallis test with Mann-Whitney U test for *post hoc* analysis **(D)** Growth curve of WT and *ulp2Δ/Δ* strains grown in non-selective (N/S) and MMS-containing liquid media. Error bars: standard deviation (SD) of three biological replicates **(E)** Growth curve of WT and *ulp2Δ/Δ* strains grown in non-selective (N/S) and HU-containing liquid media. Error bars: standard deviation (SD) of three biological replicates.

One of the highest-ranked genes on our screen is *ULP2* (CR_03820C/ *orf19.4353*: MMS score:3, UV score:3) encoding for a SUMO protease (**Fig 1A and 1B** and **S1 Table**).

Colony-forming unit (CFU) assays of UV-treated cells confirmed the importance of *C. albicans ULP2* in DNA damage resistance as UV treatment reduced the number of CFU in a *ulp2Δ/Δ* strain (~14.5% survival) compared to a wild-type (WT) strain (~33.7% survival) (**Fig 1C**). Furthermore, the *ulp2Δ/Δ* strain also displayed a reduced growth rate in liquid media containing MMS or Hydroxyurea (HU), a chemotherapeutic agent that challenges genome

integrity by stalling replication forks [57] (**Fig 1D** and **1E**). Thus, *ULP2* has a role in responding to a wide range of genotoxic agents.

## *ULP2* but not *ULP1* is required for survival under stress

*C. albicans* contains three putative SUMO-deconjugating enzymes: Ulp1, Ulp2 and Ulp3 (**Fig 2A**). Sequence comparison between the three *C. albicans* Ulp proteins and the two well-characterised *S. cerevisiae* Ulps (Ulp1 and Ulp2) reveals that although the *C. albicans* proteins are poorly conserved, the amino acid residues essential for catalytic activity are conserved (**Fig 2A** and **2B**). Accordingly, recombinantly expressed *C. albicans* Ulp1, Ulp2 and Ulp3 have SUMO-processing activity *in vitro* [58]. Similarly to *S. cerevisiae ULP1*, *C. albicans ULP3* is an essential gene and was not investigated further in this study [59,60]. Previous studies failed to detect a poly-histidine tagged Ulp2 protein by Western blot analyses of *C. albicans* protein extracts [58]. These results suggested that Ulp2 is unstable or expressed at undetectable low levels. We reassessed Ulp2 protein levels by generating strains expressing, at the endogenous locus, an epitope-tagged Ulp2 protein (Ulp2-HA). Western blot analyses show that Ulp2-HA expression is readily detected in extracts from four independent integrant strains (**Fig 2C**). Thus, a stable Ulp2 protein is expressed in cells grown under standard laboratory growth conditions (YPD, 30°C). To assess whether *ULP1*, similarly to *ULP2*, is involved in genotoxic stress response, we engineered homozygous deletion strains for *ULP1* (*ulp1Δ/Δ*) and *ULP2* (*ulp2Δ/Δ*). Growth analysis demonstrated that deletion of *ULP2* reduces fitness as the newly generated *ulp2Δ/Δ* strain is viable, but cells are slow-growing (**Fig 2D** and **2E**). In contrast, the *ulp1Δ/Δ* strain grows similarly to the WT control in solid and liquid media (**Fig 2D** and **2E**). Spot dilution assay confirmed that *ULP2* is an important regulator of *C. albicans* stress response as, similarly to the deletion library mutant, the newly generated *ulp2Δ/Δ* strain was sensitive to different stress conditions including treatment with DNA damaging agents (UV and MMS), DNA replication inhibitor (HU), oxidative stress ($H_2O_2$) and high temperature (39°C) (**Fig 2E**). In contrast, deleting *ULP1* did not cause any sensitivity to the tested stress conditions (**Fig 2E**).

Although *ULP-1*, *ULP-2* and *ULP-3* may have some partially redundant functions, our results suggest that *ULP-1* does not play a major role in genotoxic stress response. In summary, loss of *ULP2* leads to poor growth in standard laboratory growth conditions and hypersensitivity to multiple stresses.

## Genome instability is exacerbated in the absence of *ULP2*

To assess whether the hypersensitivity to DNA damage agents observed in the *ulp2Δ/Δ* strain was indeed due to enhanced genome instability, we deleted *ULP2* from a set of strains containing a heterozygous *URA3*+ marker gene inserted in three different chromosomes (Chr 1, 3 and 7) [61]. We quantified the frequency of *URA3*+ marker loss by plating on media containing the counter-selective drug 5-Fluoroorotic Acid (FOA) and scoring the number of colonies able to grow on FOA-containing media compared to non-selective (N/S) media. Deletion of *ULP2* caused a dramatic increase in LOH rate at all three chromosomes (Chr1: 378X, Chr3: 18X, Chr7: 96X), indicating that *ULP2* is required for maintaining genome stability across the *C. albicans* genome (**Fig 3A**). In *C. albicans*, hypersensitivity to genotoxic stress leads to filamentous growth [37,62–65]. Accordingly, the *ulp2Δ/Δ* strain formed wrinkled colonies on solid medium and displayed a higher frequency of abnormal morphologies than the WT strain, including filamentous cells (**Fig 3B** and **3C**). To assess whether the exacerbated *ulp2Δ/Δ* genome instability is linked to defective chromosome segregation, we deleted the *ULP2* gene in a reporter strain in which *TetO* sequences are integrated adjacent to the centromere on Chr7 (*CEN7*) and a TetR-GFP fusion protein is expressed from the gene-free *NEUT5L* locus

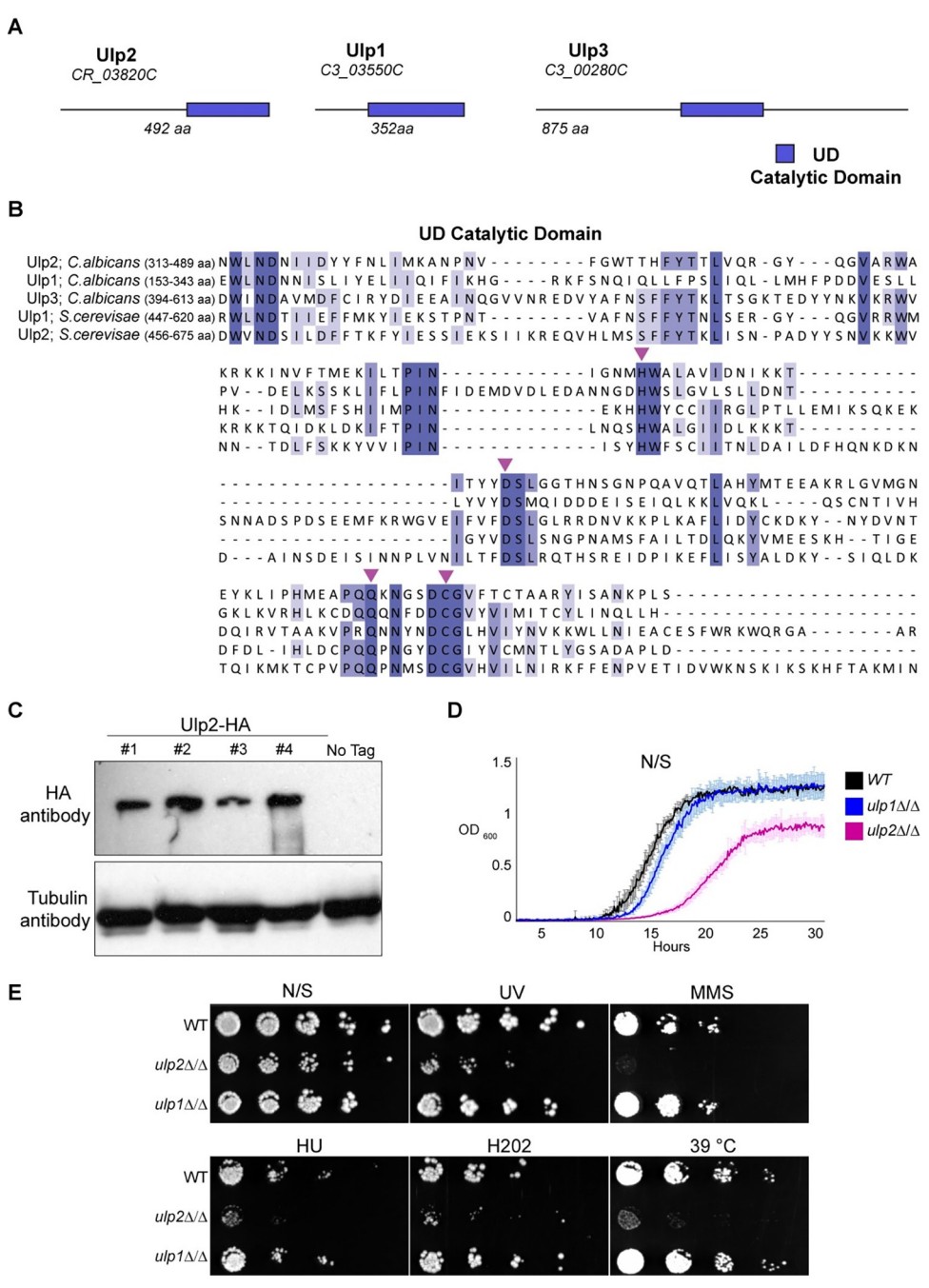

**Fig 2. *ULP2* is necessary for survival under stress. (A)** Schematic representations of *C. albicans* Ulp1, Ulp2 and Ulp3 proteins. The systematic name and the amino acid (aa) number is indicated for each protein. Blue box: putative catalytic UD SUMO protease domain **(B)** Protein alignment of *C. albicans* Ulp proteins (Ulp1, Ulp2 and Ulp3) and *S. cerevisiae* Ulp2 proteins (Ulp1 and Ulp2). Magenta arrows: amino acids essential for SUMO protease activity **(C)** Western blot analysis of 4 ULP2-HA integrants and the progenitor untagged control (No Tag). *Top*: anti-HA Western blot, *Bottom*: anti-Tubulin Western blot serving as a loading control **(D)** Growth curves of WT, *ulp1Δ/Δ* and *ulp2Δ/Δ* strains grown in non-selective (N/S) liquid media. Error bars: standard deviation (SD) of three biological replicates **(E)** Serial dilution assay of WT, *ulp1Δ/Δ* and *ulp2Δ/Δ* strains grown in unstressed (N/S) or stress (UV, MMS, HU, H2O2 and 39°C) growth conditions.

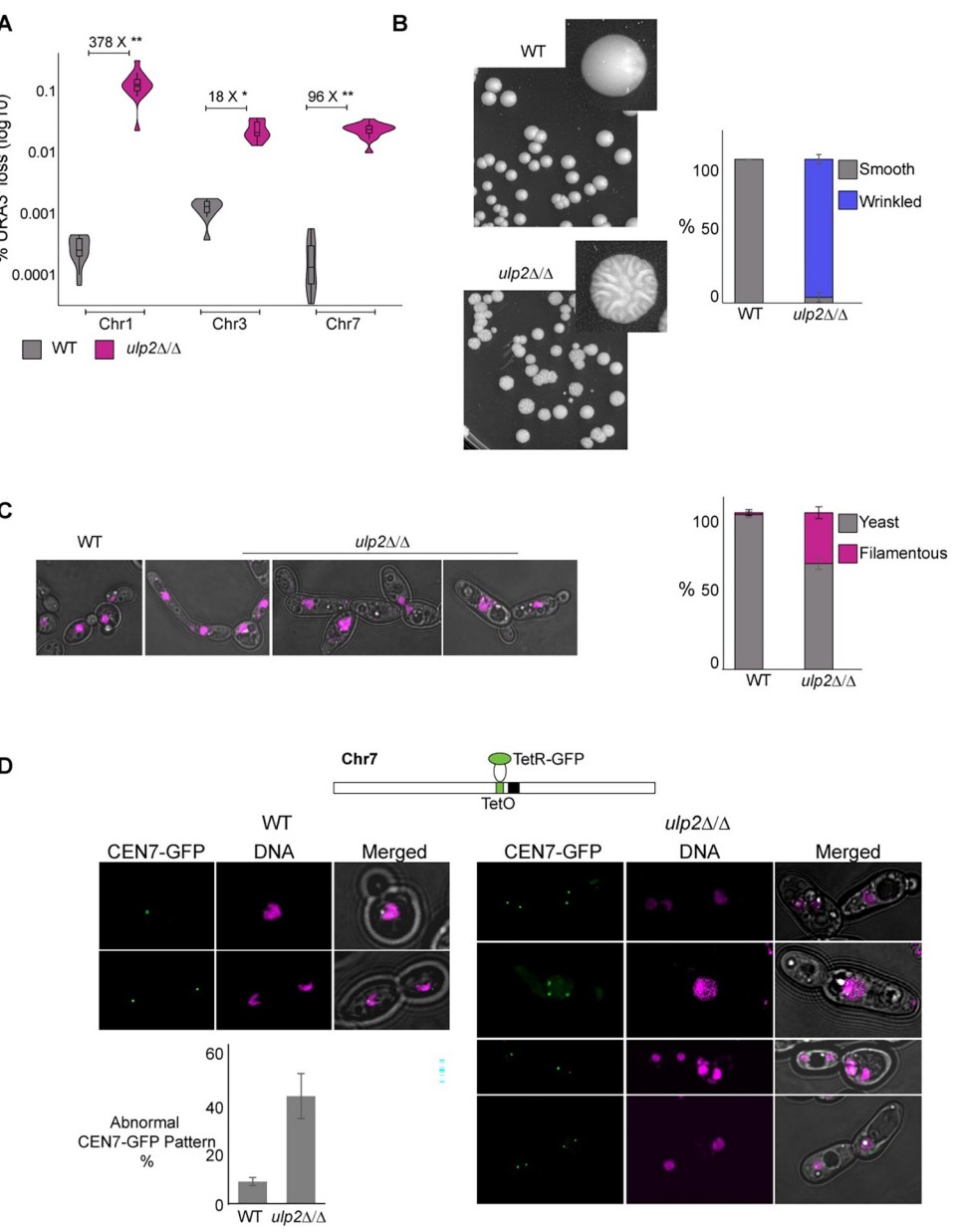

**Fig 3. Loss of *ULP2* leads to increased genome instability. (A)** Quantification (%) of loss of a heterozygous *URA3*⁺ marker gene inserted in Chr1, Chr3 and Chr7 in WT and *ulp2Δ/Δ* strain. The fold difference of *URA3*⁺ marker loss between *ulp2Δ/Δ* and WT strains is indicated. Statistical differences were calculated using the Kruskal-Wallis test and the Mann-Whitney U test for *post hoc* analysis **: Chr1 (4.11 E-07) and Chr7 (6.74 E-05) p-value, *: Chr3 (2.87 E-02) p-value **(B)** *Left*: Representative images displaying colony morphologies of WT and *ulp2Δ/Δ* strains. *Right*: Quantification (%) of smooth and wrinkled colonies in WT and *ulp2Δ/Δ* strains **(C)** *Left*: Representative images displaying the morphologies of WT and *ulp2Δ/Δ* strains. *Right*: Quantification (%) of yeast and filamentous morphologies in WT and *ulp2Δ/Δ* strains. Error bar: Standard deviation of 3 biological replicates **(D)** *Top*: schematics of the *CEN7* TetO and TetR-GFP system. *Bottom*: nuclear morphology and segregation pattern of centromere 7 (*CEN7*) in WT and *ulp2Δ/Δ* strain. Quantification (%) of abnormal GFP-CEN7 patterns is indicated. Error bar: Standard deviation of 3 biological replicates.

[66,67]. The binding of TetR-GFP to *TetO* sequences allowed the visualisation of Chr7 duplication and segregation during the cell cycle. We found that deletion of *ULP2* leads to abnormal Chr7 segregation. This included cells with no TetR-GFP signals or multiple TetR-GFP-foci,

that were ~5 fold higher in the *ulp2Δ/Δ* strain compared to the WT control strain (**Fig 3D**). Thus, deletion of *C. albicans ULP2* leads to increased genome instability.

### *ULP2* loss coupled with stress triggers selection of segmental aneuploidies

Previous studies performed in the model system *S. cerevisiae* demonstrated that loss of *ULP2* leads to the accumulation of a specific multichromosome aneuploidy (amplification of both ChrI and ChrXII). This aneuploidy rescues the lethal defects of *ulp2* deletion by amplification of specific genes on both chromosomes [68,69]. To assess whether loss of *C. albicans ULP2* triggers the selection of gross karyotypic abnormalities, we analysed the genome of WT and *ulp2Δ/Δ* strains at the beginning (Day 0) and the end (Day 30) of an *in vitro* evolution experiment where strains were passaged daily for 30 days in rich media (YPD 30°C) (**Fig 4A**). Clamped homogeneous electrical field (CHEF) electrophoresis analysis did not detect any major chromosome rearrangements in both sets of evolved strains (**Fig 4B**). To further investigate the impact of *ULP2* loss on genome organisation, we sequenced the genome of 3 randomly selected *ulp2Δ/Δ* colonies by whole genome Illumina sequencing (WGS) and compared their genome to the *C. albicans* reference genome. This analysis revealed that loss of *ULP2* leads to very few (<10 across the 3 isolates) *de novo* mutations (**S2 Table**). Although we did not detect CNVs, we identified novel LOH tracts on different chromosomes in two of the three sequenced colonies (**Fig 4C**). For example, chromosome mis-segregation followed by reduplication of the remaining homologue is detected in isolate U1 (U1: ChrR) and the

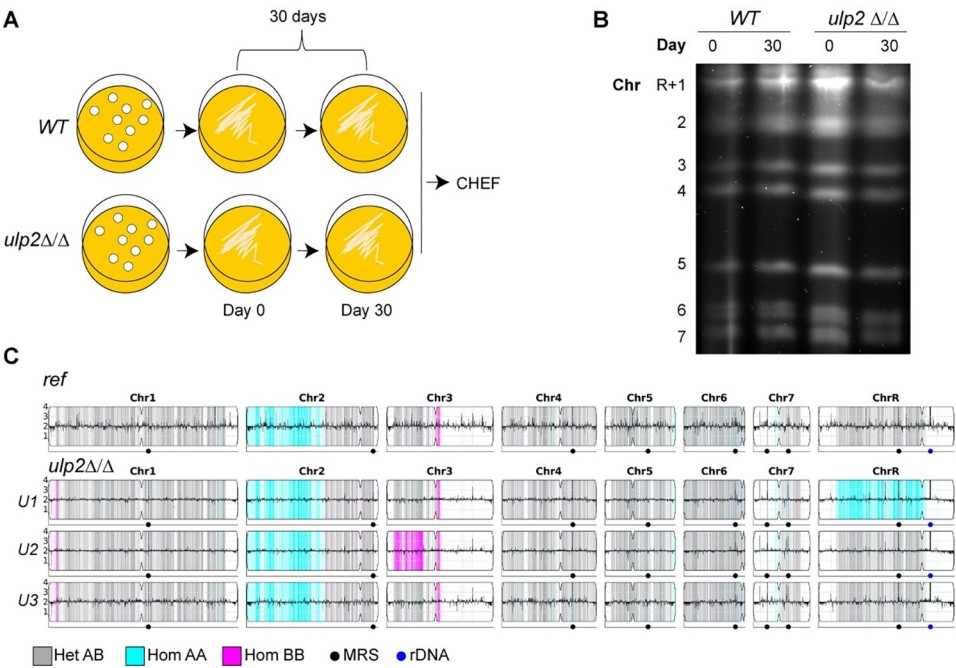

**Fig 4. Karyotypic changes are detected in the absence of *ULP2*. (A)** Schematics of laboratory evolution strategy **(B)** Karyotype organisation of *C. albicans* WT and *ulp2Δ/Δ* strains at the start (Day 0) and the end (Day 30) of the evolution experiment **(C)** Whole genome sequencing analysis of the progenitor (WT:SN152) and three single *ulp2Δ/Δ* colonies (U1, U2, and U3). Data were plotted as the log2 ratio and converted to chromosome copy number (y-axis, 1–4 copies) as a function of chromosome position (x-axis, Chr1-ChrR) using the Yeast Mapping Analysis Pipeline (YMAP) [109]. Heterozygous (AB) regions are indicated with grey shading, and homozygous regions (loss of heterozygosity) are indicated by shading of the remaining haplotype, either AA (cyan) or BB (magenta).

genome of U2 contains a long-track LOH (U2:Chr3L) that occurred within 4.6 kb of a repeat locus on Chr3L (*PGA18*, [19]) (**Fig 4C**). Therefore, loss of ULP2 can trigger selection of large chromosomal variations. We hypothesised that exposure of *ulp2Δ/Δ* cells to stress could facilitate the selection of novel adaptive karyotypes. To test this hypothesis, we challenged the *ulp2Δ/Δ* strain with high concentrations of FLC (128 μg/ml; ~1000 fold above susceptibility breakpoint [70]) and isolated a *FLC*-adapted isolate (*FLC-1*) that was still able to grow at high drug concentration following two passages (T1 and T2) in non-selective (N/S) media (**Fig 5A** and **5B**). The phenotypes associated with the loss of *ULP2* were partially rescued in *FLC-1*. as this isolate was less sensitive than the *ulp2Δ/Δ* progenitor to UV treatment and high temperature (39˚C) (**Fig 5B**). Furthermore, fewer wrinkled colonies are present in *FLC-1* than *ulp2Δ/Δ* and the number of elongated cells was reduced in *FLC-1* compared to *ulp2Δ/Δ* (**Fig 5C** and **5D**).

To identify mutations that underlie the above phenotypes, we sequenced the genome of three *FLC-1* single colonies (*FLC-1a*, *b* and *c*) by Illumina technology (**Figs 5A and S1** and **S3**–**S5 Tables).** We also sequenced the genome of 3 additional *ulp2Δ/Δ FLC*-adapted isolates (*FLC-2*, *FLC-3* and *FLC-4*) randomly selected from FLC plates and unable to grow at high FLC doses following passaging in N/S medium. The whole-genome sequencing revealed that the *FLC*-adapted colonies have a genotype distinct from the *ulp2Δ/Δ* progenitor (**Fig 5E** and **S3 Table**). *FLC-1*, but not *FLC-2*, *FLC-3*, or *FLC-4* isolates, is marked by a segmental aneuploidy: a partial Chr1 amplification (~1.3 Mbp) containing 535 protein-coding genes (**Fig 5E** and **S4 Table**). Furthermore, all sequenced isolates have a partial deletion (~ 388 Kb) of the right arm of ChrR (ChrRR-Deletion). ChrRR-deletion occurs at the ribosomal DNA and extends to the right telomere of ChrR (ChrR:1,897,750 bp—2,286,380 bp), reducing the copy number of 204 genes from two to one (**Fig 5E** and **S5 Table**). In contrast, we detected very few (<10) *de novo* point mutations, and none of these are common among all the sequenced *FLC* isolates (**S3 Table**). Thus, exposure to an antifungal drug triggers the selection of adaptive chromosomal variations in the absence of *ULP2*.

## Segmental aneuploidy is selected via a two-step process

Illumina sequencing is an inadequate technology for resolving complex chromosomal abnormalities because of the generated short reads. For example, we could not establish whether the increased Chr1 CNV was due to the formation of an extrachromosomal fragment or to a chromosomal fusion. Therefore, to understand further the genomic structure of *FLC-1* we sequenced the genome of this isolate using long-read Oxford Nanopore Technologies (ONT) sequencing. To establish the temporal trajectory of *FLC-1* aneuploidies, timepoints T1 (1X passage in non-selective media following FLC treatment) and T2 (2X passages in non-selective medium following FLC treatment) were sequenced (**Fig 5A**). Using this method, we could resolve the structure of *FLC-1* aneuploidies completely (**Fig 6A**). We discovered that *FLC-1* contains, in addition to the two endogenous Chr1 homologous chromosomes, an extra linear Chr1 (linChr1) copy that is selected in a two-step process. At time T1, we detected a linChr1 (~ 1.9 Mbp) containing an intact right arm and a truncated left arm (**Fig 6B**). At the 5' breakpoint, Chr7 subtelomeric and telomeric regions are fused to Chr1. This chromosomal fusion occurs within the internal *TLOα34* gene deleting ~1.3 Mbp. Sequence homology between *TLOα34* and the subtelomeric *TLOγ16* gene is likely to have guided the fusion between Chr1 and Chr7. At T2, linChr1 is further processed at its right arm by deletion of ~ 0.6 Mbp and the addition of telomeric repeats (**Fig 6B**). The 3' breakpoint contains a microhomology tract (6 bp 5'-TTCTTG-3') between internal sequences of Chr1 and telomeric repeats. The resulting linChr1 spans the centromere and is flanked by terminal telomeric repeats. To assess whether

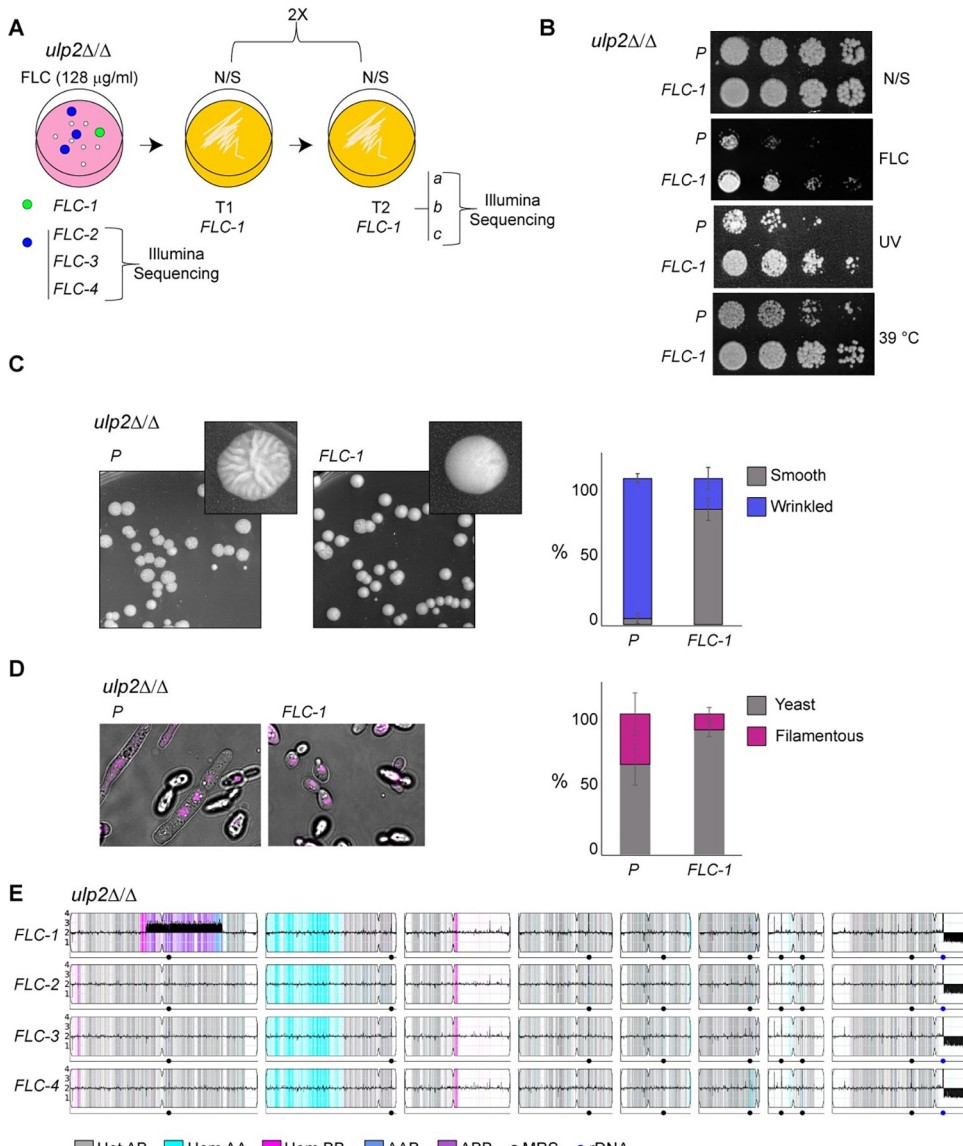

**Fig 5. Genomic variants are selected in *ulp2Δ/Δ* cells challenged with additional stress. (A)** Schematic of experimental design. The *FLC-1 ulp2Δ/Δ* isolate was selected from a casitone agar plate containing 128 μg/ml fluconazole (FLC) passaged two times (2X) non-selective (N/S) agar plates and its genome sequenced by Illumina technology. *FLC-2*, *FLC-3* and *FLC-4 ulp2Δ/Δ* isolates were selected from a casitone agar plate containing 128 μg/ml fluconazole (FLC) and the genome was sequenced by Illumina technology. **(B)** Serial dilution assay of *ulp2Δ/Δ* parental (P) and fluconazole-recovered isolates (*FLC-1*) in unstressed (N/S) or stress (UV, MMS, HU, H2O2 and 39˚C) growth conditions. **(C)** *Left*: Representative images displaying colony morphologies of *ulp2Δ/Δ* parental (P) and fluconazole-recovered isolates (*FLC-1*). *Right*: Quantification (%) of smooth and wrinkled colonies in WT and *ulp2Δ/Δ* strains **(D)** *Left*: Representative images displaying the morphologies of *ulp2Δ/Δ* parental (P) and fluconazole-recovered isolates (*FLC-1*). *Right*: Quantification (%) of yeast and filamentous morphologies in WT and *ulp2Δ/Δ* strains. Error bar: Standard deviation of 3 biological replicates **(E)** Whole genome sequencing data for four single colonies isolated from 128 μg/ml fluconazole plates (*FLC1-FLC4*). The chromosome copy number is plotted along the y-axis (1–4 copies).

linChr1 was necessary for the *ulp2Δ/Δ* phenotypic rescue, we passaged *FLC-1* in N/S media and selected three independent phenotypic revertants (*R-1*, *R-2* and *R-3*) that, similarly to the *ulp2Δ/Δ* strain, are less able to withstand FLC and form wrinkled colonies on solid media (**Fig 6C** and **6D**). Diagnostic PCR analysis with primers specific for linChr1 indicates that linChr1

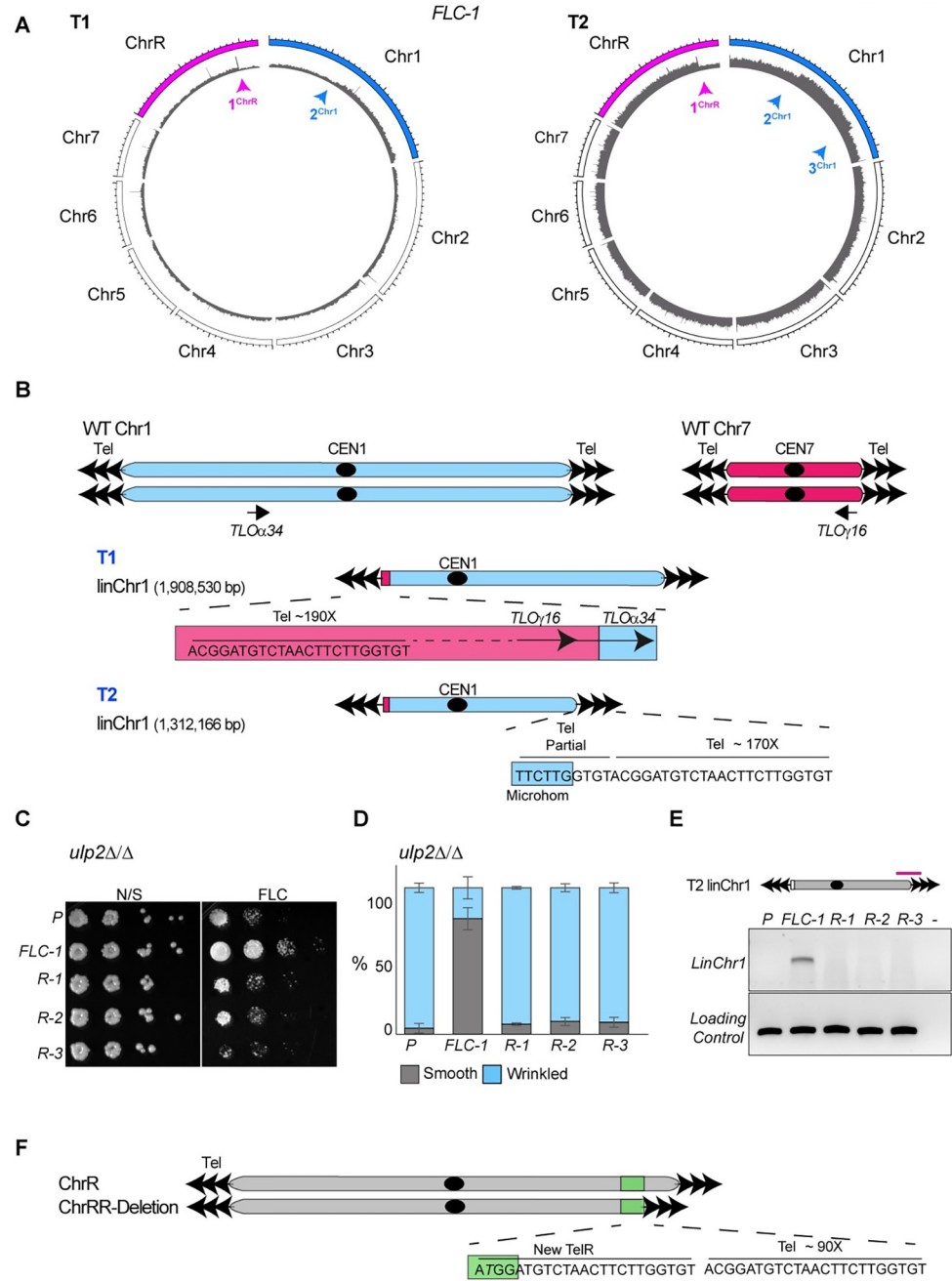

**Fig 6. Microhomology tracts and DNA repeats drive the formation of segmental aneuploidies. (A)** Circular plots of the long-read coverage across the *FLC-1* genome at timepoint T1 (left) and T2 (right). Magenta and Blue arrows: CNV breakpoints on ChrR and Chr1 respectively **(B)** Schematics of the events leading to linChr1 formation. *Top*: Schematics of the endogenous Chr1 and Chr7. *Middle*: Schematics of linChr1 structure at timepoint T1. *Bottom*: Schematics of linChr1 structure at timepoint T2. Tel: telomeric repeats. Microhom: microhomology tract found on Chr1 and telomeric repeats. **(C)** Serial dilution assay of *ulp2Δ/Δ* parental (P) fluconazole-recovered (*FLC-1*) and FLC-1 revertants (*R-1*, *R-2* and *R-3*) isolates in non-selective (N/S) agar plate or plates containing 128 μg/ml fluconazole (FLC) **(D)** Quantification (%) of smooth and wrinkled colonies in *ulp2Δ/Δ* parental (P) fluconazole-recovered (*FLC-1*) and FLC-1 revertants (*R-1*, *R-2* and *R-3*) isolates **(E)** *Top*: Schematics of linChr1 highlighting the position (magenta line) of linChr1 specific primer. *Bottom*: LinChr1 diagnostic PCR in the *ulp2Δ/Δ* parental (P), fluconazole-recovered (*FLC-1*) and FLC-1 revertants (*R-1*, *R-2* and *R-3*) isolates. Loading Control: Chr1 internal primers **(F)** Schematics of ChrRR-deletion. Tel: telomeric repeats Microhom: microhomology tract found on ChrR and telomeric repeats.

was lost in the *R-1*, *R-2* and *R-3* revertants, while an amplification product was detected in DNA isolated from *FLC-1* (**Fig 6E**). This result suggests that linChr1 has an adaptive value.

Finally, we discovered that the ChrRR-deletion is selected early following FLC treatment as this deletion is present at timepoint T1 and T2 (**Fig 6A**). A microhomology tract between the *RDN18* (encoding for the 18S rRNA) genes and telomeric repeats facilitated the addition of telomeric repeats to one copy of the *RDN18* gene stabilising the truncated chromosome (**Fig 6E**). Thus, rearrangements at microhomology regions and DNA repeats guide the formation of adaptive segmental aneuploidies via two temporally separated events.

## Discussion

There is a significant gap in our understanding of how, in microbial organisms, genome instability leads to increased fitness in stress and non-stress environments. Here, we identified the SUMO protease Ulp2 as a key protein ensuring genome stability in *C. albicans*. We demonstrated that *ULP2* loss leads to enhanced genome variation and that dysregulation of the SUMO system combined with drug treatment facilitates the selection of adaptive segmental aneuploidies via a two-step process.

### Ulp2 is a critical regulator of *C. albicans* genome stability

SUMOylation is a post-translational protein modification signalling in a large number of cellular processes by targeting nuclear eukaryotic proteins [71–74]. SUMO peptides are covalently attached to target proteins by the concerted action of E1, E2 and E3 enzymes while SUMOylation is reversed by SUMO-specific proteases [75–79]. We discovered that the SUMO protease Ulp2 promotes genome stability in *C. albicans*. These findings are consistent with the emerging role of SUMO proteases as a guardian of genome integrity across eukaryotes. Indeed, SUMO proteases ensure genome stability throughout eukaryotes [68,80]. We hypothesise that *C. albicans ULP2* promotes genome stability by modulating SUMO levels of several substrates including: *(i)* kinetochore and centromere-associated proteins, *(ii)* the DNA replication machinery and *(iii)* factors involved in DNA repair. Indeed, it is well established that SUMO homeostasis modulates kinetochore function, DNA replication and DNA repair and that defects in these pathways lead to exacerbated genome instability [77,78,81,82]. For example, several *S. cerevisiae* kinetochore and centromere-associated proteins, including the centromeric-specific histone H3 variant Cse4$^{CENP-A}$, are SUMOylated and the knockdown of SENP6, the human ortholog of Ulp2, leads to mis-localisation of the inner kinetochore complex CENP-H/I/K causing chromosome segregation defects [74,83–85]. Furthermore, SUMOylation of replication and repair proteins increases upon DNA damage [74,83,86–89]. We still know very little about SUMOylation effect on *C. albicans* biology and its adaptation to hostile host environments. However, the observation that *C. albican*s protein SUMOylation patterns are different in normal and stress growth conditions agrees with our data and suggests that this post-translation modification has a critical role in adaption [90].

### The adaptive power of segmental aneuploidy to overcome multiple stresses

It is well established that exposure to moderate stress can increase tolerance to unrelated stresses [91,92]. This increased tolerance is usually the result of coordinated gene expression changes known as the core stress response [91]. In contrast, we show that exposure to a stress (antifungal drug) can increase tolerance to unrelated stress (loss of *ULP2*) by selecting segmental aneuploidies. We demonstrate that two different segmental aneuploidies can co-exist when *ulp2*Δ/Δ cells are challenged with FLC: an amplification of a Chr1 fragment via the formation of an extra chromosome (linChr1) and a partial deletion of ChrR. We posit that these changes

in karyotype provide a synergistic fitness advantage in response to the two independent stressors (the presence of FLC and lack of *ULP2*) by simultaneously changing the copy number of multiple genes. Indeed, linChr1 amplifies 535 protein-coding genes (**S4 Table**) and GO analyses demonstrated that 41 of those genes are associated with a "response to drug" phenotype (**S6 Table**). Among these, amplification of *UPC2* encoding for the Upc2 transcription factor is likely to be critical. Indeed, it is well established that *UPC2* overexpression leads to FLC resistance by *ERG11* transcriptional upregulation [93,94]. Similarly, amplification of *CCR4* and *NOT5* (part of linChr1) might play a key role in rescuing the fitness defects of the *ulp2*Δ/Δ strain. Ccr4 and Not5 are subunits of the evolutionarily conserved Ccr4-Not complex that modulate gene expression at multiple levels, including transcription initiation, elongation, deadenylation and mRNA degradation [95]. It has been shown that *S. cerevisiae CCR4* and *NOT5* overexpression rescues the lethal defects associated with a *ulp2* deletion strain [68]. Similarly, GO analysis revealed that ChrRR-Deletion leads to a reduced copy number of 34/204 genes associated with "response to stress" and 18/204 genes are linked to "response to drug" (**S7 Table**).

## Selection of segmental aneuploidies via a two-step process involving microhomology tracts

One key question is to understand how complex novel genotypes are selected in *C. albicans*. We provide evidence that two temporally separated events led to the formation of linChr1 via regions with (micro)homology to telomeric and subtelomeric regions. Our findings support a model in which non-allelic homologous recombination (NAHR) between the *TLOα34* (Chr1) gene and Chr7 subtelomeric sequences resulted in a first deletion on Chr1 that is stabilised by the addition of Chr7 telomeric regions. We propose that, in a second step, microhomology-mediated break-induced replication (MMBIR) involving a microhomology region (6 bp 5'-TTCTTG-3') between internal sequences of Chr1 (position Chr1: 2594815–2594820 bp) and the 23 bp telomeric repeats caused the second Chr1 deletion and addition of telomeric repeats. We hypothesise that MMBIR is also responsible for the stabilisation of ChrRR-deletion. Indeed, a microhomology tract (4 bp AxGG) between the *RDN18* regions and telomeric repeats is found at ChrRR-deletion breakpoint resulting in the addition of telomeric repeats at the rDNA locus and subsequent stabilisation of the broken chromosome. As MMBIR is caused by stalled replication forks [96], it is likely that defects in DNA replication trigger MMBIR and aneuploidy selection. We suspect that these mechanistic pathways are common in *C. albicans* and other fungal pathogens. Indeed, complex aneuploidies have been observed in other stress conditions [30,97,98]. Complex novel karyotypes are most likely the result of independent events that accumulate over time. Furthermore, CNVs downstream of the rDNA locus have been described in *C. albicans* clinical isolates as well as in cells treated with the antifungal posaconazole [16,30,97]. Analysis of these genotypes by long-read sequencing will be instrumental in fully resolving these karyotypes and unveiling their origins.

## Material and methods

### Yeast strains and growth conditions

Strains used in this study are listed in **S8 Table.** Routine culturing was performed at 30˚C in Yeast Extract-Peptone-D-Glucose (YPD) liquid and solid media containing 1% yeast extract, 2% peptone, 2% dextrose, 0.1 mg/ml adenine and 0.08 mg/ml uridine, Synthetic Complete (SC-Formedium) or Casitone (5 g/L Yeast extract, 9 g/L BactoTryptone, 20 g/L Glucose, 11.5 g/L Sodium Citrate dehydrate, 15 g/L Agar) media. When indicated, media were

supplemented with 1 mg/ml 5-Fluorotic acid (5-FOA, Melford), 200 μg/ml Nourseothricin (clonNAT, Melford), 15 μg/ml and 128 μg/ml fluconazole (Sigma #F8929), 6m $H_2O_2$ (Sigma #H1009), 12 mM and 22 mM Hydroxyurea (Sigma #H8627), 0.005% MMS (Sigma #129925).

### Genetic screening

The genetic screening was performed using a *C. albicans* homozygous deletion library [36] arrayed in 96 colony format on YPD plates (145x20 mm) using a replica plater (Sigma #R2508). Control non-selective (N/S) plates were grown at 30˚C for 48 hours. UV treatment was performed using UVitec (Cambridge) with power density of 7.5 μW/cm$^2$ (0.030 J for 4 seconds). Following UV treatment, plates were incubated in the dark at 30˚C for 48 hours. For MMS treatment, the library was spotted onto YPD plates (145x20 mm) containing 0.005% MMS and incubated at 30˚C for 48 hours. UV and/or MMS sensitivity of selected strains was confirmed by serial dilution assays in control (YPD) and stress (UV: power density of 7.5 μW/cm$^2$, MMS: 0.005%) plates. Correct gene deletions were confirmed by PCR using gene-specific primers (**S9 Table**).

### Yeast strain construction

Integration and deletion of genes were performed by transforming PCR products containing a marker gene and the appropriate target-gene sequence integration site [99]. Oligonucleotides and plasmids used for strain construction are listed in **S9** and **S10 Tables**, respectively. For Lithium Acetate transformation, overnight liquid yeast cultures were diluted in fresh YPD and grown to an OD$_{600}$ of 1.3. Cells were harvested by centrifugation and washed once with d$H_2O$ and once with SORB solution (100 mM Lithium acetate, 10 mM Tris-HCl pH 7.5, 1 mM EDTA pH 7.5/8, 1M sorbitol; pH 8). The pellet was resuspended in SORB solution containing single-stranded carrier DNA (Sigma-Aldrich) and stored -80˚C in 50 μl aliquots. Frozen competent cells were defrosted on ice, mixed with 5 μL of PCR product and 300 μL PEG solution (100 mM Lithium acetate, 10 mM Tris-HCl pH 7.5, 1 mM EDTA pH 8, 40% PEG4000). Following incubation for 21–24 hours at 30˚C, cells were heat-shocked at 44˚C for 15 minutes and grown in 5 mL YPD liquid for 6 hours before plating on selective media at 30˚C.

### UV survival quantification

Following dilution of overnight liquid cultures, 500 cells were plated in YPD control plates and 1500 cells were plated in YPD stress plates and UV irradiated with a power density of 7.5 μW/cm$^2$ (0.030 J for 4 seconds). Plates were incubated at 30˚C for 48 hours in the dark. Colonies were counted using a colony counter (Stuart Scientific). Experiments were performed in 5 biological replicates, and violin plots were generated using R and R Studio IDE (http://www.r-project.org/).

### Growth curve

Overnight liquid cultures were diluted to 60 cells/μL in 100 μL YPD and incubated at 30˚C in a 96 well plate (Cellstar, #655180) with double orbital agitation of 400 rpm using a BMG Labtech SPECTROstar nanoplate reader for 48 hours. When indicated, YPD media was supplemented with MMS (0.005%) and HU (22 mM). Graphs show the mean of 3 biological replicates, error bars show the standard deviation.

### Serial dilution assay

Overnight liquid cultures were diluted to an OD$_{600}$ of 4, serially diluted 1:5 and spotted into agar plates with and without indicated additives using a replica plater (Sigma Aldrich,

#R2383). Images of the plates were taken using Syngene GBox Chemi XX6 Gel imaging system. Experiments were performed in 3 biological replicates.

## Protein extraction and Western blotting

Yeast extracts were prepared as described [100] using $1 \times 10^8$ cells from overnight cultures grown to a final $OD_{600}$ of 1.5–2. Protein extraction was performed in the presence of 2% SDS (Sigma) and 4 M acetic acid (Fisher) at 90˚C. Proteins were separated in 2% SDS (Sigma), 40% acrylamide/bis (Biorad, 161–0148) gels and transferred into PVDF membrane (Biorad) by semi-dry transfer (Biorad, Trans Blot SD, semi-dry transfer cell). Western-blot antibody detection was performed using anti-HA mouse monoclonal primary antibody (12CA5 Roche, 5 mg/ml) at a dilution of 1:1000 in PBS containing 0.2% Tween and 5% w/v non-fat dry milk, recombinant anti-alpha Tubulin (Abcam #ab184970) at a dilution of 1:10000 in PBS containing 0.2% Tween and 5% w/v non-fat dry milk, anti-mouse IgG-peroxidase (A4416 Sigma) at dilution of 1:30000, anti-rabbit IgG-peroxidase (A0545 Sigma) at a dilution of 1:30000, and Clarity ECL substrate (Bio-Rad).

## $URA3^+$ marker loss quantification

Strains were first streaked onto synthetic solid media lacking uracil and uridine (SC–Uri) to ensure the selection of cells carrying the $URA3^+$ marker gene. Parallel liquid cultures were grown for 16 hours at 30˚C in YPD and plated on SC plates containing 1 mg/ml 5-FOA (5-fluorotic acid; Sigma) and on N/S SC plates. Colonies were counted after 2 days of growth at 30˚C. The frequency of the $URA3^+$ marker loss was calculated using the formula F = m/M, where m represents the median number of colonies obtained on 5-FOA medium (corrected by the dilution factor used and the fraction of culture plated) and M the average number of colonies obtained on YPD (corrected by the dilution factor used and the fraction of culture plated) [63]. Statistical differences between results from samples were calculated using the Kruskal-Wallis test and the Mann-Whitney U test for post hoc analysis. Statistical analysis was performed and violin plots were generated using R Studio (http://www.r-project.org/).

## Microscopy

30 ml of yeast cultures ($OD_{600}$ = 1) grown in SC were centrifuged at 1550 x g for 5 minute and washed once with $dH_2O$. Cells were fixed in 10 ml of 3.7% paraformaldehyde (Sigma #F8775) for 15 minutes, washed twice with 10 ml of $KPO_4$/Sorbitol (100 mM $KPO_4$, 1.2 M Sorbitol) and resuspended in 250 µl PBS containing 10 µg of DAPI. Cells were then sonicated and resuspended in a 1% low melting point agarose (Sigma Aldrich) before mounting under a 22 mm coverslip of 0,17 µm thickness. Samples were imaged on a Zeiss LSM 880 Airyscan with a 63x/1.4NA oil objective. Airyscan images were taken with a relative pinhole diameter of 0.2 AU (airy unit) for maximal resolution and reduced noise. GFP was imaged with a 488 nm Argon laser and 495–550 nm bandpass excitation filter. The DAPI channel was imaged on a PMT with standard pinhole of 1AU and brightfield images were captured on the trans-PMT with the same excitation laser of 405 nm. DAPI and brightfield images were taken with the same pixel size and bit depth (16bit) as the airyscan images. Images were of a 42.7x42.7µm field of view with a 33 nm pixel size resolution. z-stacks were taken containing cells of z interval of 500 nm. Airyscan Veena filtering was performed with the inbuilt algorithms of Zeiss Zen Black 2.3. Experiments were performed in 3 biological replicates and >100 cells/replicate were counted.

## Drug selection

For fluconazole selection, strains were incubated overnight in Casitone liquid media at 30°C with shaking. $10^4$ cells were plated in a small plate (10 cm) containing Casitone medium plus 256μl DMSO or 128 μg/mL fluconazole. Plates were incubated at 30°C for 7 days. Colonies able to grow on fluconazole- were streaked (2X) on non-selective (N/S) plates and tested by spotting assay in Casitone +DMSO, or Casitone+FLC. For selection of *FLC-1* revertants, 100 cells were plated in YPD agar plates and single colonies were assessed for their ability to grow on casitone medium plus 256μl DMSO or 128 μg/mL fluconazole by serial dilution assays.

## Contour-clamped homogeneous electric field (CHEF) electrophoresis

Intact yeast chromosomal DNA was prepared as previously described [101]. Briefly, cells were grown overnight, and a volume equivalent to an $OD_{600}$ of 6 was washed in 50 mM EDTA and resuspended in 20 μl of 10 mg/ml Zymolyase 100T (Amsbio #120493–1) and 300 μl of 1% Low Melt agarose (Biorad # 1613112) in 100 mM EDTA. Chromosomes were separated on a 1% Megabase agarose gel (Bio-Rad) in 0.5X TBE using a CHEF DRII apparatus. Run conditions were as follows: 60-120s switch at 6 V/cm for 12 hours followed by a 120-300s switch at 4.5 V/cm for 26 hours at 14°C. The gel was stained in 0.5x TBE with ethidium bromide (0.5 μg/ml) for 60 minutes and destained in water for 30 minutes. Chromosomes were visualised using a Syngene GBox Chemi XX6 gel imaging system.

## Whole-genome Illumina sequence analysis

Illumina genome sequencing data have been deposited in the Sequence Read Archive under BioProject PRJNA781758. Genomic DNA was isolated using a phenol-chloroform extraction as previously described [26]. Paired-end (2 x 151 bp) sequencing was carried out by the Microbial Genome Sequencing Center (MiGS) on the Illumina NextSeq 2000 platform. Read trimming was conducted using Trimmomatic (v0.33 LEADING:3 Trailing:3 SLIDINGWINDOW:4:15 MINLEN:36 TOPHRED33) [102]. Trimmed reads were mapped to the *C. albicans* reference genome (SC5314 A21-s02-m09-r08) from the *Candida* Genome Database (http://www.candidagenome.org/download/sequence/C_albicans_SC5314/Assembly21/archive/C_albicans_SC5314_version_A21-s02-m09-r08_chromosomes.fasta.gz). Reads were aligned to the reference using BWA-MEM (v0.7.17) with default parameters [103]. The BAM files, containing aligned reads, were sorted and PCR duplicates removed using Samtools (v1.10 samtools sort, samtools rmdup) [104]. Qualimap (v2.2.1) analysed the BAM files for mean coverage as aligned to the SC5314 A21 reference genome; coverages ranged from 73.7x to 89.3x coverage [105]. Variant detection was conducted using the Genome Analysis Toolkit (Mutect, v2.2–25), with the SC5314 A21 reference genome as the reference fasta input, and SN152 as the normal bam input [106]. Variants were annotated using SnpEff (V4.3) [107] using the SC5314 A21 reference genome fasta and gene feature file above, and filtered using SnpSift for missense, nonsense, and synonymous mutations. Variants were verified visually using the Integrative Genomic Viewer, and variants present in SN152 were removed. (IGV, v2.8.2) [108].

## Read depth and breakpoint analysis of short-reads sequencing

Whole-genome sequencing data were analysed for copy number and allele ratio changes as previously described [19,30]. Aneuploidies were visualised using the Yeast Mapping Analysis Pipeline (YMAP, v1.0) [109]. BAM files aligned to the SC5314 reference genome as described above were uploaded to YMAP and read depth was determined and plotted as a function of

chromosome position, using the inbuilt SC5314 A21 reference genome and haplotype map. Read depth was corrected for both chromosome-end bias and GC-content. The GBrowse CNV track and allele ratio track identified regions of interest for CNV and LOH breakpoints, and more precise breakpoints were determined visually using IGV. LOH breakpoints are reported as the first and last informative homozygous position in a region that is heterozygous in the parental genome. CNV breakpoints were identified as described previously [19,30].

## Long-read DNA sequencing

Oxford Nanopore Technologies (ONT) sequencing data have been deposited in the Sequence Read Archive under BioProject PRJNA879282. DNA was extracted from an overnight culture in YPD using the QIAGEN genomic tip 100/G kit (Qiagen, #10243) according to manufacturing protocol. Long read sequencing libraries were prepared using the SQK-LSK109 Ligation Sequencing Kit with the EXP-NBD104 Native Barcoding Kit (Oxford Nanopore Technologies) from approximately 1μg of high molecular weight genomic DNA, following the manufacturer's protocol. Long read libraries were sequenced on R9.4.1 Spot-On Flow cells (FLO-MIN106) using the GridION X5 platform set to super accurate base calling.

## Long-read genome assembly

Long reads were quality controlled using NanoPlot v1.30.1 [110], and adapters and barcodes were trimmed using Porechop v0.2.4 (https://github.com/rrwick/Porechop) with default parameters. Reads shorter than 1kb or with a quality score less than Q9 were removed using Filtlong v0.2.1 https://github.com/rrwick/Filtlong. Long reads were assembled using NECAT v0.0.1_update20200803 [111] using a genome size of 16Mb, all other parameters were left as default. Error correction was performed by aligning the long reads to the assemblies with Minimap2 v2.17-r941 [112] to inform one iteration of Racon v1.4.20 [113], followed by one iteration of Medaka v1.5.0 (https://github.com/nanoporetech/medaka) using the r941_min_-high_g360 model. Assembly statistics were generated using a custom Python script, and single copy ortholog analysis was performed using BUSCO v5.2.2 [114], using the saccharomycetes_odb10 database.

## Identification of segmental aneuploidies

Raw long-reads were first aligned to the reference SC5314 genome using Minimap2 (v 2.17-r941) [112] and coverage plotted using Circos (v0.69–8) [115]. Raw reads sorting and indexing was performed with Samtools (v1.11) [116], bam to fasta conversion was conducted with bedtools (v2.30.0) [117] and visualised using IGV (v2.11.9). Presence of telomeres was confirmed by extracting raw reads at the target regions using samtools (v1.11). Individual long reads spanning the breakpoints were investigated and annotated in SnapGene Viewer.

## Supporting information

**S1 Fig. Genomic variants are selected in *ulp2Δ/Δ* cells challenged with Fluconazole. (A)** Whole genome sequence data were plotted as the log2 ratio and converted to chromosome copy number (y-axis, 1–4 copies) as a function of chromosome position (x-axis, Chr1-ChrR) using YMAP. Heterozygous (AB) regions are indicated with gray shading and homozygous regions are indicated by haplotype AA (cyan) or BB (magenta). Allele ratio changes that occur within a CNV are indicated as dark blue (AAB) or purple (ABB). Colony B and C had allele ratio colouring that was corrected using IGV and allele frequency information. **(B)** Serial dilution assay of *ulp2Δ/Δ* parental (P) and fluconazole-recovered isolates (*FLC-2*, *FLC-3*, *FLC-4*,

*FLC-1a*, *FLC-1b* and *FLC-1c*) in non-selective (N/S) or media containing 128 µg/ml fluconazole (FLC).
(PDF)

**S1 Table. Genetic screen top hits (score ≥2).**
(DOCX)

**S2 Table. List of SNPs detected in the *ulp2Δ/Δ* colonies (U1, U2 and U3) sequenced in this study.**
(XLSX)

**S3 Table. List of SNPs detected in *ulp2Δ/Δ* isolates selected from Fluconazole (128 µg/ml) plates.**
(XLSX)

**S4 Table. Coordinates of the Chromosome 1 amplification (lin-Chr1) detected in the FLC-1 *ulp2Δ/Δ* isolate.**
(XLSX)

**S5 Table. Coordinates of the Chromosome R deletion (ChrRR-Deletion) detected in FLC-1, FLC-2, FLC-3 and FLC-4 *ulp2Δ/Δ* isolates.**
(XLSX)

**S6 Table. List of genes associated with "response to drugs" GO terms for the Chromosome 1 amplification (lin-Chr1).**
(XLSX)

**S7 Table. List of genes associated with "response to stress" and "response to drugs" GO terms for the Chromosome R deletion (ChrRR-deletion).**
(XLSX)

**S8 Table. Strains used in this study.**
(DOCX)

**S9 Table. Oligonucleotides used in this study.**
(DOCX)

**S10 Table. Plasmids used in this study.**
(DOCX)

## Acknowledgments

We thank J. Berman for reagents, strains and materials and A. Pidoux for critically reading the manuscript.

## Author Contributions

**Conceptualization:** Marzia Rizzo, Anna Selmecki, Alessia Buscaino.

**Data curation:** Natthapon Soisangwan, Robert Jordan Price.

**Formal analysis:** Anna Selmecki.

**Funding acquisition:** Anna Selmecki, Alessia Buscaino.

**Investigation:** Marzia Rizzo, Natthapon Soisangwan, Samuel Vega-Estevez, Robert Jordan Price, Chloe Uyl, Elise Iracane, Matt Shaw, Jan Soetaert, Anna Selmecki, Alessia Buscaino.

**Methodology:** Marzia Rizzo, Natthapon Soisangwan, Samuel Vega-Estevez, Jan Soetaert.

**Project administration:** Alessia Buscaino.

**Resources:** Samuel Vega-Estevez.

**Supervision:** Alessia Buscaino.

**Validation:** Marzia Rizzo, Jan Soetaert.

**Visualization:** Marzia Rizzo.

**Writing – original draft:** Alessia Buscaino.

**Writing – review & editing:** Marzia Rizzo, Natthapon Soisangwan, Samuel Vega-Estevez, Robert Jordan Price, Chloe Uyl, Elise Iracane, Matt Shaw, Anna Selmecki, Alessia Buscaino.

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
