## [Decision Letter · Decision Letter 0]

14 Oct 2022

Dear Dr Buscaino,

Thank you very much for submitting your Research Article entitled 'Stress combined with loss of the Candida albicans SUMO protease Ulp2 triggers selection of aneuploidy via a two-step process' to PLOS Genetics.

The manuscript was fully evaluated at the editorial level and by independent peer reviewers. The reviewers appreciated the attention to an important problem, but raised some substantial concerns about the current manuscript. Based on the reviews, we will not be able to accept this version of the manuscript, but we would be willing to review a much-revised version. We cannot, of course, promise publication at that time.

If you decide to revise the manuscript for further consideration at PLOS Genetics, please aim to resubmit within the next 60 days, unless it will take extra time to address the concerns of the reviewers, in which case we would appreciate an expected resubmission date by email to plosgenetics@plos.org.

We are sorry that we cannot be more positive about your manuscript at this stage. Please do not hesitate to contact us if you have any concerns or questions.

Yours sincerely,

Gary P Moran

Guest Editor

PLOS Genetics

Wendy Bickmore

Section Editor

PLOS Genetics

The paper addresses the topic of genome stability and adaptation in the fungal pathogen Candida albicans and identified Ulp2 as having a role in maintaining genome stability. Loss of Ulp2 results in increased sensitivity to DNA damage and in the presence of fluconazole, genome rearrangements and fluconazole resistance.

Reviewer 1 has submitted a very positive review and has some minor suggestions for improvements.

Reviewer 2 raises some more sub substantive comments which may be addressed through changes in emphasis and experimental revisions. The reviewer is largely unconvinced of the hypothesis that loss of Ulp2 alone is insufficient to select for dramatically different genotypes and that additional stress (here fluconazole) is required to select for genotypic variants with adaptive traits. They point out that conclusive evidence for such assertions may require much larger experiments analysing large numbers of colonies. In the absence of such extensive analysis, the authors may wish to revise this hypothesis to accept that genomic variants can occur in Ulp2 strains (e.g. LOH in Fig 4) and that in the presence of stress variants can be selected with adaptive traits. Insistence that chromosomal variation only occurs in the presence of additional stress would require much larger experiments.

The reviewer also suggests examining whether this segmental duplication is stable (have the authors determined whether loss of the Chr1 element is associated with reversion to susceptibility?).

Overall, the paper brings new data on the role of Ulp2 in Candida albicans and how genome rearrangements can result in antimicrobial resistance. The approach, using a combinations of long read, short read and karyotype data to reconstruct genomic variants is hight informative and will be of interest to a general readership.

Reviewer's Responses to Questions

**Comments to the Authors:**

Reviewer #1: This article makes a substantial advance in characterizing how the SUMO protease Ulp2 influences genome stability and antifungal drug resistance in the opportunistic fungal pathogen Candida albicans. The authors employed a selective deletion library containing deletion mutants of factors important for virulence and of Candida-specific genes to screen for determinants of genome stability by testing for hypersensitity to UV radiation and the alkylating agent MMS. The screen identified 28 candidates, several were conserved factors involved in cell cycle regulation, DNA damage repair, chromosome segregation, or stress response; these are obvious hits and demonstrate that the setup of the screen is working. More importantly, there were 3 candidates with no obvious homology in the main yeast models (Saccharomyces cerevisiae and Schizosaccharomyces pombe). One of the best hits (high sensitivity to UV and MMS) was ULP2 which codes for a SUMO protease. SUMOylation is a post-translational modification which has been reported to be involved in response to environmental changes.

Candida albicans contains three ULP genes (ULP1-3). The encoded proteins are poorly conserved in S. cerevisiae with the exception for their catalytic motifs. C. albicans ULP3 is essential. Hence, the authors made independent deletions of ULP1 and ULP2 (it is good practice to not reuse strains from the deletion library) for further characterization. As corroboration of the screen the ulp2 mutant showed reduced fitness and hypersensitivity to UV, MMS, hydroxyurea, and hydrogen peroxide. The ulp1 mutant was indiscernible from wildtype, and its characterization was not further pursued.

Using genetic and cytological marker-loss assays the authors established that the ulp2 mutant displayed increased genome instability. 3 randomly selected ulp2 mutant colonies were whole-genome sequenced, and signs of Loss-of-Heterozygosity (LoH) were detected. However, since no gross chromosome rearrangements were observed in these strains, the authors addressed the possibility that loss of ULP2 in combination with stress would trigger larger genome reorganisations. This was tested using the antifungal fluconazole (FLC). Indeed, colonies adapted to high drug concentrations arose more readily in the ulp2 mutant than in wildtype. 4 FLC-adapted colonies were then picked at random and whole-genome sequenced using an Illumina platform. To resolve potential copy number variations and chromosome rearrangements better, one of the FLC-adapted isolates FLC-1 was also whole-genome sequenced using Oxford Nanopore technology. This enabled the dissection of the chromosome aberrancies in FLC-1. The authors then suggest a two-step process to explain the genome reorganisations they observed; this is perfectly sensible.

The data is of high quality and supports the authors’ conclusions, I have only a few minor issues I have detected.

Please ensure that your BioProjects are public, currently they cannot be accessed.

Specific Comments:

- L.150: please spell out genus names of the two model yeasts, it’s their first mention.

- L.163: surely, this should refer to Fig 1C.

- L.184-198: Can you exclude that the function of Ulp1 is not redundant with one, or both, of the other Ulps? Would it be worthwhile making at least the ulp1 ulp2 double mutant to check?

- L.220: Revise “included including”.

- L.289: “detected a linChr1”.

- L.296: space missing between “Chr1 and”

- L.353: there’s a single floating quotation mark.

- L.424, L.428: It’s TRIS-HCl not HCL

- Refs. 7, 10, 16, 19, 20, 21, 24, 25, 29, 30, 31, 44, 52, 61, 69, 70, 73, 74, 79, 80, 87, 90, 91, 95, 96, 97, 102, 105, 108, 110, 116 have issues (missing volume numbers, incomplete page range, missing article ID, page count instead of article, and/or obviously wrong weblink)

Reviewer #2: In this paper, the authors delete Candida albicans ULP2 encoding SUMO deconjugation enzyme and analyze the consequences of loss of SUMOylation, including genome stability and its role in adaptation to antifungal fluconazole. The authors show that loss of ULP2 causes chromosome instability, which, when combined with drug selective pressure, can lead to aneuploidy that, possibly, confers drug resistance and rescues some other phenotypic consequences of ULP2 loss.

Major points:

It has been previously established that the lack of ULP2 in Saccharomyces cerevisiae is associated with slow growth, abnormal spindles, irregularly shaped colonies, higher rates of chromosomal loss, and sensitivity to UV, gamma rays, heat, MMS, and hydroxyurea. This study, thus, merely confirms S. cerevisiae ULP2 properties in the opportunistic pathogen C. albicans.

Lines 239 to 241, the authors say that no whole-chromosome aneuploidies were found in ulp2��� strain. However, in lines 243 to 244 (and Fig. 4C), they report chromosome 1 missegregation followed by reduplication of the remaining homologue in strain U1. Chromosome instability/aneuploidy in strains lacking ULP2 has to be addressed with a larger number of clones, not just three of them.

Lines 247 to 249 – The hypothesis that the stress associated with loss of ULP2 is insufficient to select for dramatically different genotypes is unsubstantiated. Authors need to address more thoroughly the question whether the lack of ULP2 alone triggers chromosome aneuploidies in C. albicans similarly to S. cerevisiae. Please, do CHEF and WGS with large number of both independent ulp2��� clones and independent FLC clones. Present WGS profiles of all clones in the same figure.

Adaptation to fluconazole needs a more systematic study. Please, obtain more clones in independent experiments, conduct a standard broth microdilution assay to determine the minimum inhibitory concentration (MIC) in FLC1, 2, 3, 4, etc clones. Visualize segmental aneuploidies with CHEF.

To address whether the segmental duplication in FLC1 causes fluconazole resistance, the authors have to prepare independent phenotypic revertants to fluconazole sensitivity and to determine whether the segmental duplication is lost.

Overall, while adaptation to fluconazole due to segmental aneuploidies needs more studies, a portion of this work characterizing the properties of C. albicans ULP2 could be published in a more specialized yeast journal.

Minor points:

Line 251 – What was the reason for using a high concentration, 128 μg/ml, of fluconazole?

Numerous typos have to be checked through the manuscript

**Have all data underlying the figures and results presented in the manuscript been provided?**

Reviewer #1: Yes

Reviewer #2: Yes

PLOS authors have the option to publish the peer review history of their article (what does this mean?). If published, this will include your full peer review and any attached files.

Reviewer #1: No

Reviewer #2: No

---

## [Editor Report · Decision Letter 1]

16 Dec 2022

Dear Dr Buscaino,

We are pleased to inform you that your manuscript entitled "Stress combined with loss of the Candida albicans SUMO protease Ulp2 triggers selection of aneuploidy via a two-step process" has been editorially accepted for publication in PLOS Genetics. Congratulations!

Yours sincerely,

Gary P Moran

Guest Editor

PLOS Genetics

Wendy Bickmore

Section Editor

PLOS Genetics

Comments from the reviewers (if applicable):

Dear Dr Buscaino,

Thank you for submitting a revised version of your manuscript "Stress combined with loss of the Candida albicans SUMO protease Ulp2 triggers selection of aneuploidy via a two-step process". I am happy to see that you have revised your hypothesis following consideration of the reviewers' comments. Recognition that loss of ULP2 leads to the formation of genomic variants and that exposure to additional stress leads to the selection of novel karyotypes with adaptive potential, fits well with the data described. I am particularly encouraged to see that additional experiments have been performed demonstrating that loss of fluconazole resistance is associated with loss of the chromosome 1 amplification. The additional minor corrections are welcome and improve clarity. For this reason I am recommending acceptance of the manuscript in its revised form.

**Data Deposition**

http://datadryad.org/submit?journalID=pgenetics&manu=PGENETICS-D-22-01058R1

**Press Queries**

---

## [Editor Report · Acceptance letter]

21 Dec 2022

PGENETICS-D-22-01058R1 

Stress combined with loss of the Candida albicans SUMO protease Ulp2 triggers selection of aneuploidy via a two-step process 

Dear Dr Buscaino, 

We are pleased to inform you that your manuscript entitled "Stress combined with loss of the Candida albicans SUMO protease Ulp2 triggers selection of aneuploidy via a two-step process" has been formally accepted for publication in PLOS Genetics! Your manuscript is now with our production department and you will be notified of the publication date in due course.

With kind regards,

Anita Estes

PLOS Genetics

On behalf of:
